# Development and Evaluation of a Keypoint-Based Video Stabilization Pipeline for Oral Capillaroscopy

**DOI:** 10.3390/s25185738

**Published:** 2025-09-15

**Authors:** Vito Gentile, Vincenzo Taormina, Luana Conte, Giorgio De Nunzio, Giuseppe Raso, Donato Cascio

**Affiliations:** 1Department of Physics and Chemistry “E. Segre”, University of Palermo, 90128 Palermo, Italy; vito.gentile@synbrain.ai (V.G.); luana.conte@unipa.it (L.C.); giuseppe.raso@unipa.it (G.R.); 2synbrAIn S.r.l., Viale Regione Siciliana Nord-Ovest 8639, 90147 Palermo, Italy; 3Department of Mathematics and Informatics, University of Palermo, 90128 Palermo, Italy; vincenzo.taormina@unipa.it; 4Laboratory of Biomedical Physics and Environment, Department of Mathematics and Physics “E. De Giorgi”, University of Salento, 73100 Lecce, Italy; giorgio.denunzio@unisalento.it; 5National Institute for Nuclear Physics (INFN), 73100 Lecce, Italy; 6Laboratory of Advanced Data Analysis for Medicine (ADAM), Interdisciplinary Research Applied to Medicine, University of Salento and Local Health Authority of Lecce, 73100 Lecce, Italy

**Keywords:** capillaroscopy, microcirculation, video stabilization, signal enhancement, capillaries segmentation, SIFT, ORB, GFTT

## Abstract

Capillaroscopy imaging is a non-invasive technique used to examine the microcirculation of the oral mucosa. However, the acquired video sequences are often affected by motion noise and shaking, which can compromise diagnostic accuracy and hinder the development of automated systems for capillary identification and segmentation. To address these challenges, we implemented a comprehensive video stabilization model, structured as a multi-phase pipeline and visually represented through a flow-chart. The proposed method integrates keypoint extraction, optical flow estimation, and affine transformation-based frame alignment to enhance video stability. Within this framework, we evaluated the performance of three keypoint extraction algorithms—Scale-Invariant Feature Transform (SIFT), Oriented FAST and Rotated BRIEF (ORB) and Good Features to Track (GFTT)—on a curated dataset of oral capillaroscopy videos. To simulate real-world acquisition conditions, synthetic tremors were introduced via Gaussian affine transformations. Experimental results demonstrate that all three algorithms yield comparable stabilization performance, with GFTT offering slightly higher structural fidelity and ORB excelling in computational efficiency. These findings validate the effectiveness of the proposed model and highlight its potential for improving the quality and reliability of oral videocapillaroscopy imaging. Experimental evaluation showed that the proposed pipeline achieved an average SSIM of 0.789 and reduced jitter to 25.8, compared to the perturbed input sequences. In addition, path smoothness and RMS errors (translation and rotation) consistently indicated improved stabilization across all tested feature extractors. Compared to previous stabilization approaches in nailfold capillaroscopy, our method achieved comparable or superior structural fidelity while maintaining computational efficiency.

## 1. Introduction

Capillaroscopy imaging is a non-invasive technique used to examine the microcirculation of various body parts, including the oral mucosa [1]. This approach has been instrumental in diagnosing and monitoring diseases that affect the microvasculature [1]. Morphological evaluation of capillaries can provide useful insights into the diagnosis of several pathologies [2,3]. However, capillaroscopic images often suffer from motion artifacts and shaking, which compromise diagnostic accuracy [1,4]. These effects are critical not only for visual assessment by physicians but also for the development of computer-assisted systems for the detection, segmentation [5], and evaluation of pathological capillaries [6].

A capillaroscopic video captures sequential frames of a specific region in the oral cavity over a given time span. Between consecutive frames (e.g., *n* and n+1), significant misalignments may arise due to probe motion (translation and rotation) or patient-related involuntary movements such as breathing. An example of this misalignment is shown in Figure 1.

Another challenge in the stabilization process stems from the dynamic displacement of red blood cells (RBCs). In larger capillaries, the continuous flow of RBCs yields a nearly seamless vessel representation. In finer capillaries, however, RBC movement may appear intermittent, producing discontinuities between frames and a stroke-like visualization of the vessels.

In computer vision, video stabilization techniques aim to mitigate the effects of undesired motion by estimating and compensating for frame-to-frame displacements [7]. In our context, stabilization seeks to align consecutive frames with a focus on the vascular structures, while disregarding transient objects such as isolated RBCs. Post-stabilization, however, these RBCs become fundamental for reconstructing the finest microvascular details.

Video stabilization algorithms can be used to correct these issues, but their performance depends on the choice of keypoint extraction method [8,9]. The use of different keypoint extraction algorithms, such as SIFT [10], ORB [11], SURF [12], or GFTT [13], has been explored in various contexts, but their application in the field of capillaroscopy imaging is still an open research question [8,9].

The stabilization of oral capillaroscopy videos represents a crucial step toward enabling reliable quantitative analysis of the microcirculation. While previous studies have mainly focused on nailfold capillaroscopy or generic medical imaging, only limited attention has been devoted to the oral district, where non-rigid perturbations and probe-induced artifacts make stabilization particularly challenging. This methodological gap hinders the development of automated pipelines for microvascular segmentation and, ultimately, for clinical applications.

In light of these motivations, the main contributions of this work are as follows:1.First application of a structured stabilization pipeline to oral capillaroscopy, addressing the unique characteristics and motion artifacts of this imaging modality.2.Construction of a reproducible synthetic tremor dataset to simulate probe-induced perturbations and enable quantitative comparison of methods.3.Integration of keypoint extraction, sparse optical flow, affine transformations, and cumulative alignment into a modular framework, ensuring transparency, replicability, and adaptability.4.Comprehensive evaluation using multiple quantitative metrics (SSIM, Jitter Index, Path Smoothness, RMS errors) combined with qualitative visual validation.

The remainder of this paper is organized as follows: Section 2 reviews related works, Section 3 describes the proposed methodology, Section 4 reports experimental results, Section 5 discusses the findings, and Section 6 concludes the paper with future directions.

## 2. Related Works

Capillaroscopy imaging is a non-invasive technique used to examine the microcirculation of various body parts, including the oral mucosa [6,14]. It has proven instrumental in diagnosing and monitoring diseases that affect the microvasculature, particularly in systemic autoimmune conditions [14]. Quantitative analysis of capillaroscopic images, including automated approaches, holds the potential to enhance diagnostic accuracy and reproducibility.

Despite its clinical utility, capillaroscopy suffers from several technical challenges. One of the main issues is the presence of motion artifacts, which arise due to patient movement or unintentional camera shifts during acquisition [14]. These disturbances compromise image clarity and may lead to diagnostic uncertainty. In video sequences that lack well-defined feature points, the use of traditional feature-based motion estimation can result in low matching accuracy, leading to suboptimal stabilization outcomes [15].

Video stabilization is therefore a crucial enhancement technique, aimed at reducing or eliminating unwanted motion to produce visually stable sequences. It plays a key role not only in clinical applications but also in consumer-grade video content creation, which has seen exponential growth on platforms such as YouTube and Instagram [9]. Various stabilization strategies have emerged over time, ranging from traditional two-dimensional (2D) and three-dimensional (3D) methods to hybrid 2.5D approaches. The L1 optimal camera path method introduced by Grundmann et al. is a classic example of 2D stabilization, while Liu et al. pioneered a content-preserving 3D stabilization technique. Wang et al. later extended these concepts to a hybrid-dimensional (2.5D) approach, incorporating subspace constraints to model camera motion more accurately [9,15,16].

To the best of our knowledge, no prior studies have specifically addressed the problem of video stabilization in the context of oral capillaroscopy. Our work seeks to bridge this gap by evaluating the performance of three widely used keypoint extraction algorithms—Scale-Invariant Feature Transform (SIFT) [10], Oriented fast and Rotated Brief (ORB) [11], and Good Features To Track (GFTT) [13]—for stabilizing capillaroscopic video sequences acquired from the oral mucosa.

While the application of stabilization techniques to oral capillaroscopy remains unexplored, some efforts have been made in related domains. In particular, Kim et al. [17] proposed an automatic method for detecting and counting white blood cells in nailfold capillaroscopy videos. Their approach integrated a stabilization process focused solely on translational motion, deliberately excluding rotation. It relied on maximizing the cross-correlation between consecutive frames in the frequency domain, using fast Fourier transforms (FFTs) for motion estimation and compensation.

Similarly, Wang C. [18] compared two stabilization strategies—block matching and SURF feature-based matching—for nailfold video sequences. After extensive testing, block matching was selected for implementation due to its assumption of static frames without rotation or deformation. Interestingly, although the SURF method showed superior matching accuracy due to its scale and rotation invariance, its computational complexity made it less favorable for practical use in this specific application.

A broader overview of video stabilization techniques was provided by Wang et al. in a comprehensive survey [15]. The authors categorized existing methods into conventional and learning-based techniques. Notably, deep learning approaches demonstrated strong performance in handling complex camera motions and maintaining structural integrity. The survey also discussed the evaluation of stabilization performance using public datasets—such as HUJ, NUS, QMUL, Selfie, and DeepStab—based on metrics including cropping ratio, distortion value, stability score, and accumulated optical flow. Future research directions emphasized the need for perceptual evaluation metrics that account for lighting, scene semantics, and background consistency.

In another structured review, Guilluy et al. [19] addressed the absence of standardized evaluation procedures in digital video stabilization (DVS). They proposed an incremental framework that decomposes the stabilization pipeline into key stages: analysis, correction, and post-processing. The authors highlighted the necessity of unified benchmarks and evaluation protocols to ensure reproducibility and comparability among stabilization methods. They also stressed the importance of accounting for application-specific challenges, such as varying scene depth or low texture in medical imaging.

Earlier work by Balakirsky and Chellappa [20] explored the relevance of video stabilization in medical imaging, particularly in nailfold videocapillaroscopy. They noted that patient movement during acquisition can introduce artifacts that hinder both qualitative and quantitative analysis. Mechanical stabilization techniques (e.g., metal braces) have traditionally been used to mitigate such motion. However, the authors argued that digital stabilization offers a more adaptable and potentially robust alternative, relying on sequential processes including keypoint detection, motion estimation, and compensation.

Further developments in feature-based stabilization are exemplified by Raj et al. [21], who proposed a method based on boosted HAAR cascades and representative point matching. Their results demonstrated superior performance compared to traditional approaches, achieving improvements in inter-frame transformation fidelity (ITF) and modest gains in average processing time (APT). This method was particularly effective in video sequences dominated by human facial features, offering relevance for future applications in oral imaging.

In light of these findings, our current study goes beyond the isolated evaluation of keypoint extraction algorithms by implementing a structured and integrated video stabilization model tailored for oral capillaroscopy. This model, detailed in the flow-chart presented in Section 3, encompasses multiple stages—from keypoint detection and optical flow estimation to affine transformation and frame alignment—forming a coherent pipeline designed to mitigate motion artifacts in medical imaging. Within this framework, we systematically assess the performance of three widely adopted keypoint extraction methods—SIFT, ORB, and GFTT—using a curated dataset of oral capillaroscopy videos augmented with synthetic tremors. The evaluation is based on both structural fidelity and computational efficiency, providing insights into the trade-offs between accuracy and speed. Our results demonstrate that this preprocessing pipeline significantly enhances video stability, thereby laying the groundwork for future development of automated diagnostic tools based on capillaroscopic imagery of the oral mucosa [6]. Moreover, the modularity and generalizability of the proposed model may be of interest to the broader research community working on biomedical video analysis and healthcare-oriented computer vision [22].

## 3. Materials and Methods

In response to the challenges presented by motion noise and vibration in oral capillaroscopy imaging, a comprehensive methodology was developed, encompassing keypoint extraction algorithms, optical flow methods, and advanced video stabilization techniques. A schematic overview of the proposed methodology is presented in Figure 2. This flow-chart outlines the main steps of the stabilization pipeline, including keypoint extraction, motion estimation, and frame alignment. It serves as a visual guide to the detailed descriptions provided in the following subsections.

Each phase of this method was meticulously designed to address specific aspects of motion-related issues encountered during image acquisition and processing. The subsequent subparagraphs will delve into the intricacies of each phase, providing insights into the methodologies employed and their contributions to mitigating the effects of motion noise and vibration in oral capillaroscopy imaging.

To ensure clarity and reproducibility, the following subsections are organized to reflect the step-by-step sequence illustrated in Figure 2. Each block in the diagram corresponds to a specific methodological phase. This structure ensures that each component of the pipeline is explained in detail and in the same order as presented in the flow-chart.

### 3.1. Dataset Description

In this work, the database presented in [6] was used. The videos were acquired with the Horus video-capillaroscope (Adamo srl, Trapani, Italy). An optical or digital probe is connected to the central unit through a 2-meter-long fiber bundle. This setup allows for data transmission between the probe and the central unit. Additionally, there is a high-resolution color micro-television camera equipped with a high-magnification zoom lens system, capable of magnification up to 500 times. This camera captures detailed images for analysis. The central unit is paired with a high-resolution personal computer with a dedicated graphics card, which enhances the system capabilities for image processing and analysis.

Furthermore, there is a real-time analog to DV converter that links the central unit to a secondary personal computer. This secondary computer is used specifically for recording images at a rapid rate of 120 frames per second. The recorded images have a spatial resolution of 640 × 480 pixels and are in 8-bit grayscale format, providing detailed data for further examination. Due to the need for manual positioning of the probe, which involves contact with the interior of the mouth, excessively high magnifications were avoided. These high magnifications led to significant instability in the resulting videos, making it impossible to accurately track blood flow. Consequently, the operator acquired videos for this study using magnifications of 150×. In order to quantitatively evaluate a stabilization method, a dataset that includes pairs of unstabilized and stabilized videos is needed. To this end, a curated set of stable video segments was selected from the oral capillaroscopy database described in [6]. These segments were identified by three independent researchers and used to construct a golden dataset for evaluation. To ensure robustness in the evaluation, we generated synthetic unstable versions per video using controlled affine transformations. A representative stable video segment from the curated dataset is shown in Figure 3, illustrating the typical quality and resolution of the acquired sequences. This video serves as a reference for the artificially perturbed versions used in the evaluation.

### 3.2. Simulation of Tremors

To rigorously evaluate the effectiveness of the proposed video stabilization approach, it was essential to generate unstabilized video sequences that realistically mimic the motion disturbances encountered during clinical acquisition. Since the original dataset consisted of stable capillaroscopy recordings, synthetic tremors were introduced to simulate real-world acquisition conditions characterized by involuntary patient movements and probe instability.

Each frame was perturbed using an affine transformation that combines translation, rotation, and scaling. The affine transformation matrix $T_{i}$ applied to frame *i* is defined as:$$T_{i}=\left[\begin{array}{ccc} s\cdot \cos\theta & -s\cdot \sin\theta & \Delta x\\ s\cdot \sin\theta & s\cdot \cos\theta & \Delta y \end{array}\right]$$
where *s* is the scaling factor, θ is the rotation angle (in radians), and Δx, Δy are the horizontal and vertical translations, respectively.

To this end, a Gaussian noise model with zero mean was employed to generate random affine transformations for each frame. These transformations included small perturbations in translation, rotation, and scale, thereby replicating the typical motion noise observed in handheld imaging systems. The degree of perturbation was controlled by adjusting the variance of the Gaussian distribution, allowing for the simulation of different levels of tremor intensity.

Each transformation parameter was independently sampled from a Gaussian distribution, as follows:Translation (Δx, Δy): sampled from $N(0,\sigma_{t}^{2})$. Values are normalized with respect to image width and height.Rotation (θ): sampled from $N(0,\sigma_{r}^{2})$.Scaling (*s*): sampled from $N(1,\sigma_{s}^{2})$.

The same variance range [0.005, 0.01] was applied to translation, rotation, and scaling parameters to simulate comparable relative perturbations across different transformation types. This normalization ensures that each type of motion artifact contributes similarly to the overall instability, despite their differing units. These ranges were selected to mimic the typical amplitude of motion artifacts encountered during oral capillaroscopy, such as involuntary patient movements or probe instability. The normalization ensures that the perturbations are proportional to the image scale, and the use of Gaussian distributions reflects the stochastic and non-uniform nature of clinical motion.

Each frame in the stable video sequence was independently transformed using these randomly generated affine matrices, resulting in a set of artificially destabilized videos. This controlled simulation environment enabled a systematic and reproducible evaluation of the stabilization pipeline under varying motion conditions.

The variance parameters ($\sigma_{t}^{2},\sigma_{r}^{2},\sigma_{s}^{2}$) were systematically varied to generate multiple levels of tremor intensity. These levels were later used to analyze the performance of the stabilization pipeline in terms of SSIM, jitter, and geometric accuracy.

By applying the same stabilization algorithm to these perturbed sequences, we were able to assess the robustness and adaptability of the method across a spectrum of motion artifacts. This approach provided valuable insights into the algorithm’s performance in realistic clinical scenarios, where motion-induced distortions are often unpredictable and non-uniform.

It is important to note that the acquisition system used in this study does not introduce significant nonlinear lens distortions. The optical setup is calibrated and optimized for close-range imaging, and the probe design ensures minimal geometric deformation. Therefore, no correction for lens distortion was deemed necessary in the stabilization pipeline.

### 3.3. Keypoint Extraction Algorithms

Three distinct keypoint extraction algorithms—SIFT [10], ORB [11], and GFTT [13]—were selected for evaluation due to their complementary characteristics and established effectiveness in computer vision applications. SIFT is robust to scale and rotation, ORB provides a good balance between computational efficiency and accuracy, and GFTT is well-suited for detecting corner-like features, which are prevalent in capillaroscopic imagery. These methods were chosen to represent a diverse set of approaches relevant to keypoint-based video stabilization. No pre-processing steps such as contrast enhancement or denoising were applied, in order to assess the raw performance of each algorithm under realistic acquisition conditions.

SIFT (Scale-Invariant Feature Transform) is a classical algorithm known for its robustness to changes in scale, rotation, and illumination. It detects keypoints by identifying extrema in the Difference of Gaussians (DoG) across multiple scales and assigns orientations based on local image gradients. This makes SIFT particularly effective in scenarios where capillary structures may appear at different magnifications or orientations due to probe movement. Despite its accuracy, SIFT is computationally intensive, which can be a limitation in real-time applications.

ORB (Oriented FAST and Rotated BRIEF) is a more recent algorithm designed to be both fast and efficient. It combines the FAST keypoint detector with the BRIEF descriptor, adding orientation compensation to achieve rotation invariance. ORB is significantly faster than SIFT and is well-suited for real-time processing, making it a practical choice for clinical environments where computational resources may be limited. However, it may be less robust in highly textured or low-contrast regions.

GFTT (Good Features to Track), introduced by Shi and Tomasi, is a corner detection method that identifies points in the image with strong intensity variation in all directions. It is particularly effective for tracking motion across frames, which is essential in video stabilization. GFTT is computationally lightweight and performs well in scenarios with subtle or smooth motion, such as the slow flow of red blood cells in capillaries. Additionally, the decision was made to explore other algorithms, albeit briefly, including SURF [12]. It is noteworthy that SURF, a robust algorithm similar to SIFT but faster, was excluded from our evaluation due to patent considerations. Indeed, SURF cannot be included, for instance, in open source libraries such as OpenCV, thus limiting its application in several scenarios. For us, this was a sufficient reason to exclude it from our evaluation.

Apart from the latter consideration, the rationale behind the algorithm selection was to assess a diverse set of methods, each with its unique characteristics, to identify the most suitable approach for capillaroscopy video stabilization. Figure 4 illustrates the top keypoints detected in two consecutive frames, each marked with distinct colors to facilitate visual matching. This representation helps highlight the effectiveness of the keypoint extraction algorithms in identifying salient features across frames.

### 3.4. Optical Flow Algorithms

In conjunction with keypoint extraction, optical flow algorithms played a pivotal role in estimating motion between consecutive frames. Among the various approaches available, the algorithm proposed by Lucas–Kanade [23] was selected due to its favorable balance between computational efficiency and accuracy in capturing subtle motion patterns, particularly in medical imaging contexts. Its suitability for real-time applications and robustness to local intensity variations further motivated its adoption.

The Lucas–Kanade method is a differential technique that assumes brightness constancy and small motion between frames. It estimates motion vectors by solving a system of linear equations derived from spatial and temporal image gradients, assuming local constancy within a small window. This makes it effective in scenarios with smooth and continuous motion, such as capillaroscopic sequences.

This constraint is mathematically expressed as:(1)$$I_{x}u+I_{y}v+I_{t}=0$$
where $I_{x},I_{y},I_{t}$ are the spatial and temporal image gradients, and u,v are the components of the optical flow vector.

In our implementation, the algorithm was applied in its sparse variant, where motion vectors are estimated only at selected keypoints rather than at every pixel. Sparse flow was preferred as we are tracking stable keypoints rather than estimating a full dense field, which reduced computational cost and increased robustness to local noise. Keypoints were detected using feature extraction methods such as SIFT, ORB, or GFTT, which identify salient and stable regions of the image suitable for tracking across frames. This choice ensured robustness to local image perturbations while significantly reducing computational cost compared to dense methods.

Parameter tuning was performed implicitly by relying on the optimized default settings of the OpenCV implementation of Lucas–Kanade. This configuration, which is known to provide a robust balance between accuracy and computational cost, included a window size of 21 × 21 pixels, three pyramid levels, and a termination criterion based on a maximum of 30 iterations or an epsilon threshold of 0.01.

Alternative methods such as Horn–Schunck [24] were considered; however, Lucas–Kanade was preferred due to its local window-based estimation, which offers greater robustness to noise and better adaptability to non-rigid motion in capillaroscopic sequences.

### 3.5. Affine Transformation and Frame Warping

Building upon the results obtained from the previous phases of keypoint extraction and optical flow analysis, the video stabilization technique was designed as a structured pipeline composed of sequential steps. Each step contributes to progressively reducing motion artifacts and enhancing the visual coherence of the video sequence. The methodology is outlined as follows:Affine Transformation Matrix Computation: For each pair of consecutive frames, an affine transformation matrix was computed based on the estimated motion vectors derived from the optical flow analysis. This matrix encodes translation, rotation, and scaling parameters that describe the inter-frame motion.Inverse Affine Transformation and Frame Alignment: The computed affine matrix was then inverted and applied to the current frame to align it with the previous one. This transformation compensates for motion-induced distortions and ensures spatial consistency across frames.Cumulative Transformation Update: To maintain global alignment throughout the video sequence, each new transformation matrix was multiplied with the cumulative transformation of the preceding frames. This iterative process ensures that all frames are consistently aligned with the initial reference frame, minimizing drift and preserving structural integrity.

The affine transformation matrix $T_{i}$ estimated between frames can be expressed as:(2)$$T_{i}=\left[\begin{array}{ccc} s\cos\theta & -s\sin\theta & \Delta x\\ s\sin\theta & s\cos\theta & \Delta y \end{array}\right]$$
where *s* is the scaling factor, θ the rotation angle, and Δx, Δy the translation components along the horizontal and vertical axes, respectively.

The inverse transformation applied to frame i+1 is given by:(3)$$F_{i+1}^{\prime }=T_{i}^{-1}\cdot F_{i+1}$$

To maintain global consistency across the sequence, a cumulative adjustment was performed by chaining the inverse transformations:(4)$$C_{i+1}=T_{1}^{-1}\cdot T_{2}^{-1}\cdot \ldots \cdot T_{i}^{-1}$$

Each frame was then warped using the cumulative matrix $C_{i+1}$, ensuring alignment with the reference frame and minimizing drift over time.

To apply the transformations, bilinear interpolation was used to resample pixel intensities, ensuring smooth transitions and minimizing artifacts. Border regions introduced by the warping process were handled using reflection padding, which preserves edge continuity and avoids artificial gradients.

This stabilization strategy enables the correction of both global and local motion artifacts, resulting in smoother and more analyzable video sequences. The approach is particularly well-suited for capillaroscopy, where even minor misalignments can hinder the accurate visualization of microvascular structures. Moreover, this framework provides a robust basis for evaluating the impact of different keypoint extraction algorithms on the overall stabilization performance.

### 3.6. Evaluation Metrics

To quantitatively assess the performance of the stabilization pipeline, we employed a set of complementary evaluation metrics that capture structural, temporal, and geometric aspects of video quality. The Structural Similarity Index (SSIM) is a widely used metric for assessing image quality based on the similarity of their structures compared to a reference image [25,26]. In simple terms, SSIM compares two images to evaluate how similar they are in terms of their structural features such as contrast, brightness, and texture.

In the context of evaluating video stabilization methods, SSIM can be used to assess how much the stabilization method has altered or maintained the image quality during the stabilization process. A significant reduction in SSIM may indicate a loss of detail or distortion in the image due to stabilization.(5)$$\mathrm{SSIM}\left(x,y\right)=\frac{\left(2\mu_{x}\mu_{y}+c_{1}\right)\left(2\sigma_{xy}+c_{2}\right)}{\left(\mu_{x}^{2}+\mu_{y}^{2}+c_{1}\right)\left(\sigma_{x}^{2}+\sigma_{y}^{2}+c_{2}\right)}$$
where:*x* and *y* are the two images being compared.$\mu_{x}$ and $\mu_{y}$ are the means of the pixel intensities in images *x* and *y*, respectively.$\sigma_{x}$ and $\sigma_{y}$ are the standard deviations of the pixel intensities in images *x* and *y*, respectively.$\sigma_{xy}$ is the covariance between the pixel intensities in images *x* and *y*.$c_{1}$ and $c_{2}$ are two stabilized constants to prevent division by zero.

The SSIM result is a value ranging from −1 to 1, where 1 indicates perfect structural similarity between the two images.

To evaluate the effectiveness of video stabilization methods, SSIM has been calculated between the original image and the stabilized image. A value closer to 1 indicates that the image quality has been preserved during stabilization, while a lower value may indicate a reduction in image quality.

In addition to SSIM, which provides a structural similarity measure, we introduced complementary temporal and geometric metrics to better capture stabilization performance. The following metrics were used:Jitter Index: This metric quantifies frame-to-frame motion variability. It is computed as the standard deviation of inter-frame displacement vectors. Lower values indicate smoother transitions and better temporal stability.Path Smoothness Index: This index reflects the regularity of the motion trajectory across frames. A lower value indicates a more stable and coherent motion path, which is desirable in medical imaging.Geometric RMS Errors: To assess geometric fidelity, we compute the Root Mean Square (RMS) errors in translation (dx, dy) and rotation, comparing the recovered affine parameters with the ground-truth perturbations. Lower RMS values indicate more accurate motion compensation.Execution Time: To evaluate computational efficiency, we record the average execution time per video for each algorithm. This metric is crucial for assessing the feasibility of real-time deployment.

### 3.7. Benchmark Platform

To ensure the practical applicability of the proposed stabilization pipeline, all experiments were conducted on a standard desktop configuration without GPU acceleration. This setup reflects the typical computational environment available to medical practitioners and researchers, particularly in clinical or academic settings where high-performance hardware may not be accessible. By avoiding specialized hardware, we aimed to demonstrate the feasibility of our approach under realistic constraints, thereby enhancing the generalizability of our findings.

The benchmark platform consisted of a quad-core CPU with 16 GB of RAM, running a Linux-based operating system. All algorithms were implemented using Python version 3.12.4. and OpenCV libraries, ensuring reproducibility and compatibility with widely adopted open source tools.

This controlled simulation framework allowed us to assess the performance of each algorithm across varying degrees of motion noise. By maintaining consistency in the perturbation parameters, we ensured a fair comparison between methods and facilitated a detailed analysis of their stabilization capabilities under challenging conditions.

## 4. Results

The results of the analysis conducted are reported in Table 1.

The results, summarized in Table 1, show that the three algorithms (SIFT, ORB, and GFTT) perform very similarly in terms of structural similarity (SSIM), with only marginal differences. GFTT achieved the highest average SSIM score (0.7897), followed closely by ORB (0.7894) and SIFT (0.7893). The standard deviation of SSIM was also comparable among the methods, indicating consistent performance across the tested video sequences.

Beyond SSIM, temporal and geometric metrics provide additional insight into stabilization performance. The Jitter Index (≈25.9) and Path Smoothness Index (≈0.0416) were comparable across all algorithms, confirming that motion irregularities were consistently reduced. Geometric fidelity was also maintained, with RMS translation errors below 8 pixels (dx) and 6 pixels (dy), and RMS rotation errors around 1.28–1.29°, across all methods.

Standard deviations were within 4–5% for SSIM and about 25–35% for temporal and geometric indices, indicating moderate variability across sequences.

Figure 5 shows the comparison of the three algorithms in terms of average SSIM and execution time. While GFTT slightly outperforms the others in terms of SSIM, ORB demonstrates the fastest execution time, making it suitable for real-time applications.

To complement the quantitative evaluation, Figure 6 presents a side-by-side animated comparison between a perturbed input video and its corresponding stabilized output. This visual example highlights the effectiveness of the proposed pipeline in reducing motion artifacts and improving structural coherence across frames.

### Timings

Computational efficiency is a crucial aspect of video stabilization algorithms, especially in real-time applications. Timings for each algorithm were recorded to assess the computational overhead introduced during stabilization. The recorded average execution time was 0.3529 s for GFTT, 0.1857 s for ORB (best), and 1.0493 s for SIFT (worst). These differences are consistent with the computational complexity of the algorithms, with ORB remaining the most suitable choice for real-time applications, while GFTT and SIFT offer comparable accuracy at higher computational cost.

## 5. Discussions

In this study, we proposed and implemented a structured video stabilization model specifically designed for oral capillaroscopy imaging. The model, articulated in a modular pipeline and illustrated through a dedicated flow-chart, integrates keypoint extraction, optical flow estimation, and affine transformation-based frame alignment. This comprehensive approach addresses the challenges posed by motion artifacts in capillaroscopic video sequences, which are critical for both clinical interpretation and the development of automated diagnostic tools [27,28,29].

To evaluate the effectiveness of the stabilization process, we employed the Structural Similarity Index (SSIM) as a quantitative metric across video sequences perturbed by synthetic tremors. Rather than focusing on the individual characteristics of SIFT, ORB, and GFTT—which are well-established in the literature—we emphasize the novelty of integrating these components into a unified pipeline tailored to biomedical imaging.

From a performance standpoint, ORB emerged as the most computationally efficient algorithm, significantly reducing processing time compared to SIFT and GFTT. This characteristic makes ORB a compelling choice for real-time or resource-constrained environments, such as point-of-care systems or embedded medical devices.

These findings underscore a fundamental trade-off between accuracy and speed: GFTT offers marginally better visual fidelity, while ORB ensures faster execution. The choice between them should be guided by the specific requirements of the clinical or research context—whether the priority is diagnostic precision or operational responsiveness.

Importantly, this work represents one of the first implementations of a complete video stabilization pipeline tailored to oral capillaroscopy.

Compared to previous stabilization approaches in nailfold capillaroscopy, such as FFT-based cross-correlation [17] and block matching [18], our method supports affine transformations and sparse optical flow, offering greater flexibility and robustness. Block matching assumes static scenes and lacks rotation compensation, while FFT-based registration is limited to translational motion. Our pipeline, by contrast, is modular and adaptable to complex motion patterns typical of oral imaging.

By adapting and evaluating classical computer vision techniques within this specialized domain, our study contributes a reproducible and extensible framework that can serve as a foundation for future research in biomedical video analysis.

Building on this foundation, future work may explore hybrid strategies that combine the strengths of GFTT and ORB, or leverage deep learning-based motion estimation to enhance robustness in the presence of complex, non-linear artifacts. Additionally, the development of domain-specific perceptual metrics could provide more clinically meaningful assessments of stabilization quality, beyond traditional structural similarity measures.

## 6. Conclusions

This study presents a modular and reproducible video stabilization pipeline specifically designed for oral capillaroscopy imaging. The proposed method integrates keypoint extraction, sparse optical flow estimation, and affine transformation-based frame alignment, addressing the unique challenges of motion artifacts in biomedical video sequences. The experimental results demonstrate that:The pipeline effectively stabilizes videos affected by synthetic tremors, improving visual coherence.ORB is the most computationally efficient algorithm, making it suitable for real-time or embedded applications.GFTT offers slightly better structural fidelity, which may be preferable in diagnostic contexts.

Despite these promising results, the study has some limitations:The evaluation was conducted on a synthetically perturbed dataset; real-world variability may introduce additional challenges.The metrics used (e.g., SSIM, Jitter Index) provide structural and geometric insights but may not fully capture clinical perceptual quality.

While the current evaluation relies on synthetically perturbed sequences derived from stable recordings, future work will include experiments on real-world unstabilized acquisitions. This will allow us to disentangle the effects of probe-induced motion from physiological dynamics such as blood flow, and to assess stabilization quality under more variable clinical conditions.

Future work will focus on:Integrating deep learning-based motion estimation to handle non-linear and complex artifacts.Defining domain-specific perceptual metrics that reflect clinical relevance.Validating the pipeline on larger and more diverse datasets, including real-world acquisitions from multiple clinical centers.We plan to include direct experimental comparisons with classical stabilization methods (e.g., Horn–Schunck) to further benchmark the performance of our pipeline and validate its advantages in terms of structural fidelity and computational efficiency.

Overall, this work lays the foundation for robust and adaptable video stabilization in oral capillaroscopy, contributing to the advancement of automated diagnostic tools in microvascular analysis.

## Figures and Tables

**Figure 1 sensors-25-05738-f001:**
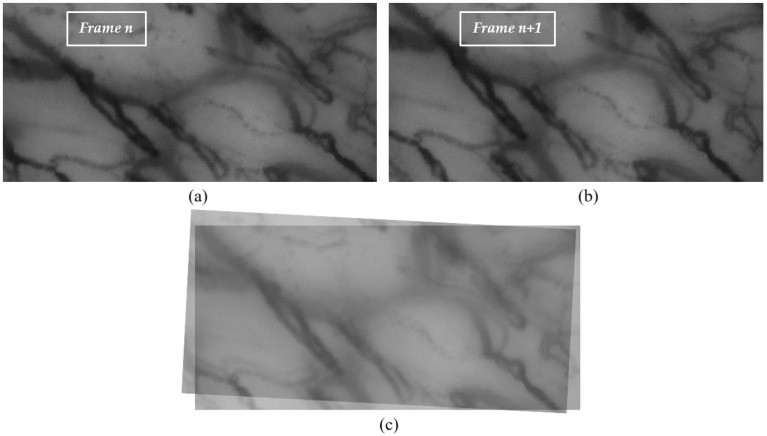
Illustration of the transition between sequential frames captured using the capillaroscope, focusing on the lower lip region. (**a**) Frame *n*; (**b**) Frame *n* + 1; (**c**) Overlap with transparency between the two consecutive frames, highlighting the roto-translation that occurred during acquisition.

**Figure 2 sensors-25-05738-f002:**
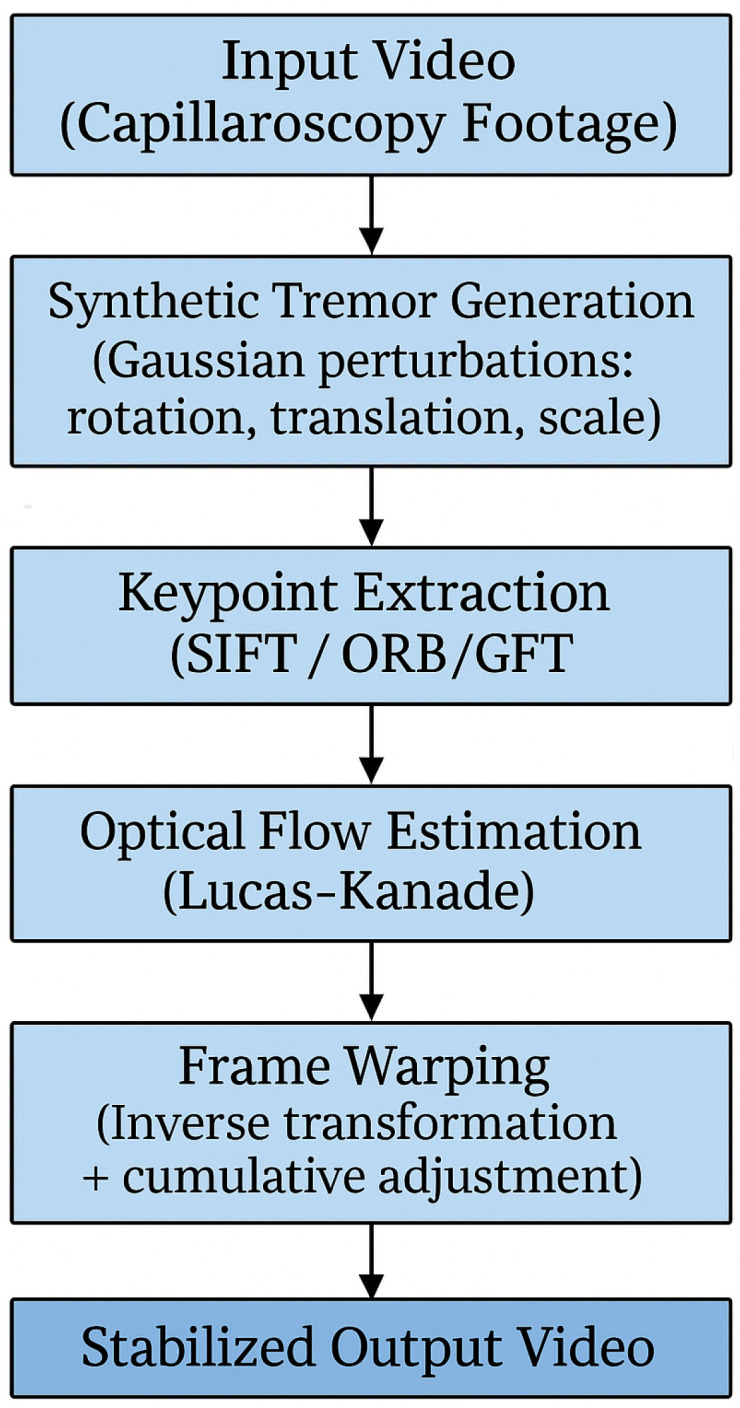
Overview of the video stabilization pipeline applied to capillaroscopy footage. The process begins with the input video, followed by synthetic tremor generation using Gaussian perturbations (rotation, translation, scale). Keypoints are extracted using feature detectors such as SIFT, ORB, or GFTT. Inter-frame motion is estimated using the Lucas–Kanade optical flow algorithm and modeled with affine transformations. Finally, frame warping is applied using inverse transformations and cumulative adjustments to produce a stabilized output video.

**Figure 3 sensors-25-05738-f003:**
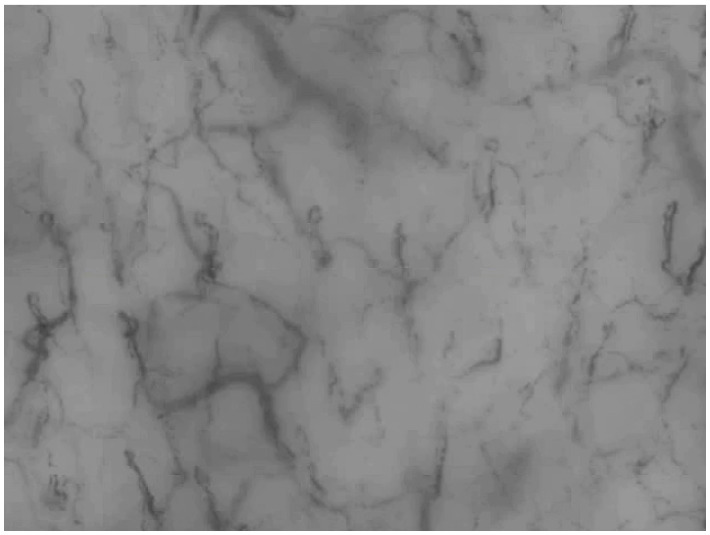
Representative stable video segment from the curated oral capillaroscopy dataset. It serves as a visual reference for the artificially perturbed versions used in the evaluation (While the static image shown here does not capture the effect, the temporal dynamics are fully appreciable in the animated sequence provided online: https://unipa-my.sharepoint.com/:v:/g/personal/donato_cascio_unipa_it/ERb68BUgz61DosuY9MYRHQIB78N9Tk93IG-3aWKPsmY2kg?e=vfLOVv (accessed on 10 September 2025)).

**Figure 4 sensors-25-05738-f004:**
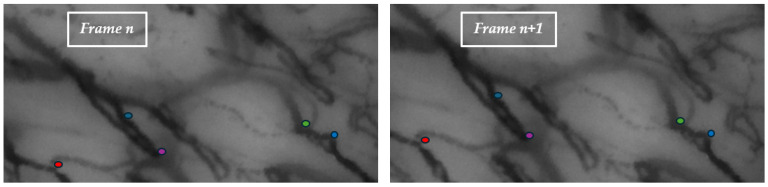
The five best keypoints found in two consecutive frames, marked with distinct colors to simplify visual matching.

**Figure 5 sensors-25-05738-f005:**
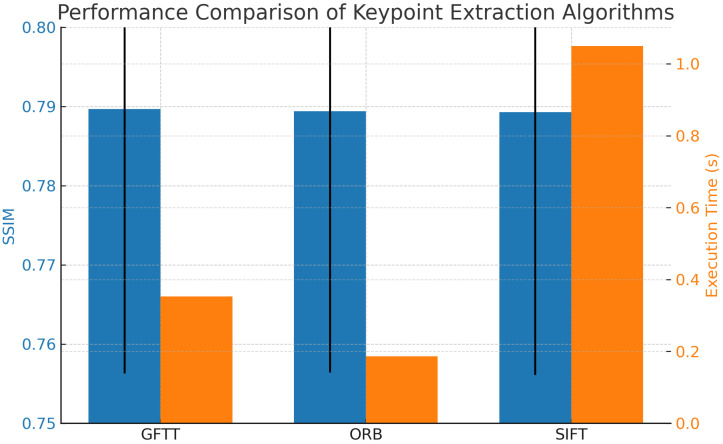
Comparison of average SSIM and execution time for each keypoint extraction algorithm.

**Figure 6 sensors-25-05738-f006:**
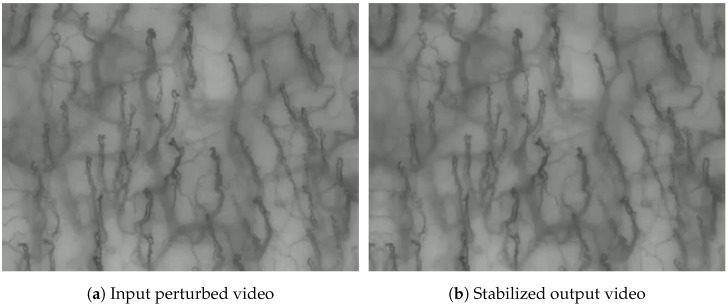
Side-by-side comparison of input (perturbed) and output (stabilized) video segments. Quantitative evaluation confirms the improvement: the Structural Similarity Index (SSIM) increased from 0.713 (perturbed) to 0.788 (stabilized), while the Jitter Index decreased from 21.81 to 1.16. These results are consistent with the visual evidence and support the effectiveness of the proposed stabilization pipeline (The dynamic behavior cannot be appreciated in the static version shown here; the full animated sequence is provided online: https://unipa-my.sharepoint.com/:v:/g/personal/donato_cascio_unipa_it/Edaaqv9GZn9PsCo-21tW0FYB2US2RcRAPowy08fUXpN7fQ?e=5BoXye (accessed on 10 September 2025)).

**Table 1 sensors-25-05738-t001:** Mean values of structural, temporal, and geometric metrics for video stabilization performance.

	SSIM	Jitter Index	Path Smoothness	RMS dx	RMS dy	RMS Rotation
GFTT	0.7897	25.84	0.0417	7.31	5.55	1.29
ORB	0.7894	25.90	0.0416	7.32	5.58	1.28
SIFT	0.7893	25.87	0.0416	7.32	5.54	1.28

## Data Availability

Data are contained within the article.

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
