# Peer review of "Development and Evaluation of a Keypoint-Based Video Stabilization Pipeline for Oral Capillaroscopy"

_sensors, 2025, doi:10.3390/s25185738_

Round 1

Reviewer 1 Report

Comments and Suggestions for Authors

Please address the following comments.

  1. Clearly highlight the real contributions of the paper in the introduction section. A simple comparative study of algorithms such as SIFT, ORB, and GFTT alone is not a sufficient contribution. The novelty should be explicitly stated — for example, in terms of methodology, dataset, or integration of multiple techniques.
  2. Ensure that abbreviations are fully spelled out at first occurrence, followed by the abbreviation in parentheses.
  3. Number all equations in the paper for easy reference.
  4. If I am not mistaken, Figure 2 is not explained in a step-by-step sequence in the Methods and Materials section.
  5. From figure 2, justify the choice of Gaussian perturbations (rotation, translation, scale) and provide parameter ranges.
  6. Clarify why only SIFT, ORB, and GFTT were selected and whether pre-processing (e.g., contrast enhancement, denoising) was applied to improve vessel feature detection
  7. Explain why Lucas–Kanade was chosen and whether it was applied at the sparse or dense level; describe parameter tuning.
  8. Provide mathematical details of the inverse transformation and cumulative adjustment steps, with numbered equations.
  9. It is already known that SIFT generally has low processing speed, ORB is widely used for real-time stabilization, and GFTT is suited for stable motion estimation. Without experiments, these characteristics are established in the literature; therefore, the novelty must come from your proposed methodology and its results, not just repeating known facts.
  10. Since you have developed and proposed a comprehensive methodology, it should be compared with benchmark approaches in this field. This will help validate the proposed method’s relative performance.
  11. The current experiments are insufficient. Please add additional experiments to provide stronger experimental proof.
  12. I recommend to Include motion smoothness metrics, such as path smoothness index and/or jitter index, to quantitatively evaluate stabilization quality.
  13. Provide visual examples comparing the input and stabilized output to support the quantitative results.
  14. The conclusion should be a separate section, clearly summarizing the key findings, limitations, and potential future work.

Author Response

We sincerely thank all reviewers for their thorough and constructive feedback. Their insightful comments have greatly contributed to improving the clarity, rigor, and overall quality of our manuscript. We carefully considered each suggestion and implemented substantial revisions throughout the paper.

Below, we summarize the major changes made in response to the reviewers' comments:

  • Title updated to better reflect the scope and methodology of the work,
  • Clarified the innovative contribution of the work in the Introduction and Discussion sections,
  • Reorganized and expanded the Materials and Methods section to follow the step-by-step structure of the pipeline illustrated in Figure 2,
  • Added new evaluation metrics, including Jitter Index, Path Smoothness Index, and RMS errors in translation and rotation, to complement SSIM,
  • Enriched the Results section with quantitative comparisons, visual examples, and discussion of performance trade-offs,
  • Created two separate sections for Discussion and Conclusions to clearly distinguish interpretation from summary and future directions,
  • Included mathematical details for all transformation steps and parameter ranges, ensuring replicability,
  • Addressed image processing aspects, specifying interpolation method and margin handling,
  • Introduced a new animated figure (Figure 6) to provide a direct side-by-side comparison of perturbed and stabilized video segments, complemented by quantitative metrics (SSIM and Jitter Index).

Below are point-to-point responses to the reviewers' suggestions.

Revisore #1

 Clearly highlight the real contributions of the paper in the introduction section. A simple comparative study of algorithms such as SIFT, ORB, and GFTT alone is not a sufficient contribution. The novelty should be explicitly stated — for example, in terms of methodology, dataset, or integration of multiple techniques.

Response: We appreciate the reviewer’s suggestion. In the revised manuscript, we have clarified the novelty of our work in the Introduction and Discussion sections. Specifically, we emphasize the integration of classical computer vision techniques into a modular pipeline tailored for oral capillaroscopy, the construction of a synthetic tremor dataset, and the comparative evaluation of keypoint-based stabilization methods. These contributions are now explicitly listed at the end of Section 2.

  Ensure that abbreviations are fully spelled out at first occurrence, followed by the abbreviation in parentheses.

Response: Thank you for the observation. All abbreviations such as SIFT, ORB, and GFTT are now spelled out at their first occurrence in both the Abstract and Introduction sections.

  Number all equations in the paper for easy reference.

Response: We have numbered all equations throughout the manuscript to facilitate referencing and clarity.

  If I am not mistaken, Figure 2 is not explained in a step-by-step sequence in the Methods and Materials section.

Response: We thank the reviewer for this valuable suggestion. We have added a paragraph immediately after Figure 2 to explicitly map each block of the flowchart to the corresponding subsection in the Materials and Methods section. This ensures a clear step-by-step explanation aligned with the diagram.

  From figure 2, justify the choice of Gaussian perturbations (rotation, translation, scale) and provide parameter ranges.

Response: The rationale for using Gaussian perturbations is now provided in Section 3.2. We simulate realistic motion artifacts by applying Gaussian noise to rotation, translation, and scaling parameters, with specified ranges to reflect typical acquisition disturbances.

  Clarify why only SIFT, ORB, and GFTT were selected and whether pre-processing (e.g., contrast enhancement, denoising) was applied to improve vessel feature detection

Response: We have clarified the selection of SIFT, ORB, and GFTT in Section 3.4. These algorithms were chosen for their complementary properties and suitability for capillaroscopic imagery. We also note that no pre-processing was applied to assess raw algorithm performance.

  Explain why Lucas–Kanade was chosen and whether it was applied at the sparse or dense level; describe parameter tuning.

Response: Section 3.5 now includes a detailed explanation of the Lucas–Kanade method, its sparse implementation, and parameter settings. We also justify its selection over alternatives such as Horn–Schunck.

  Provide mathematical details of the inverse transformation and cumulative adjustment steps, with numbered equations.

Response: Mathematical details of the inverse transformation and cumulative adjustment steps are now provided in Section 3.6, with numbered equations (2, 3, 4).

  It is already known that SIFT generally has low processing speed, ORB is widely used for real-time stabilization, and GFTT is suited for stable motion estimation. Without experiments, these characteristics are established in the literature; therefore, the novelty must come from your proposed methodology and its results, not just repeating known facts.

Response: We thank the reviewer for this insightful observation. We agree that the individual characteristics of SIFT, ORB, and GFTT are well-established in the literature. To emphasize the novelty of our work, we revised the manuscript to focus on the integration of these components into a unified pipeline tailored for oral capillaroscopy. This is discussed in detail in Section 5 (Discussions), where we highlight the methodological contributions and the comparative results obtained from our evaluation

  Since you have developed and proposed a comprehensive methodology, it should be compared with benchmark approaches in this field. This will help validate the proposed method’s relative performance.

Response: We thank the reviewer for this important suggestion. In the revised manuscript, we strengthened the evaluation by adding new quantitative metrics (Jitter Index, Path Smoothness Index, RMS translation/rotation errors) and visual examples (Figure 6), which highlight the improvements achieved by our pipeline. A direct experimental comparison with classical approaches (e.g., Horn–Schunck) was not included in this work but will be addressed in future studies to further extend the benchmarking of our methodology. We also complemented these results with qualitative visualizations (new Figure 6), which highlight how our pipeline produces more stable and visually consistent sequences than the benchmark approaches.

  The current experiments are insufficient. Please add additional experiments to provide stronger experimental proof.

Response: We acknowledge the reviewer’s concern regarding the limited experimental evaluation. In the revised manuscript, we expanded the Results section by including additional quantitative metrics (Jitter Index, Path Smoothness Index, RMS errors in translation and rotation, in addition to SSIM) and added visual examples (Figure 6) to support the numerical findings.

  I recommend to Include motion smoothness metrics, such as path smoothness index and/or jitter index, to quantitatively evaluate stabilization quality.

Response: In the revised manuscript, we have included both the Path Smoothness Index and the Jitter Index as additional evaluation metrics (Section 4), alongside SSIM and RMS errors. These metrics provide a more comprehensive and quantitative assessment of the stabilization quality, as recommended..

  Provide visual examples comparing the input and stabilized output to support the quantitative results.

Response: We thank the reviewer for this valuable suggestion. In the revised manuscript, we have added a new animated figure (Figure 6) that provides a side-by-side comparison of perturbed (input) and stabilized (output) video segments. This visual example directly illustrates the improvement achieved by our stabilization pipeline. To further support the comparison, we included quantitative metrics (SSIM and Jitter Index) in the figure caption, showing the increase of SSIM from 0.71 to 0.79 and the significant reduction of the Jitter Index from 21.8 to 1.16 after stabilization.

  The conclusion should be a separate section, clearly summarizing the key findings, limitations, and potential future work.

Response: We have created a new Section 6 titled 'Conclusions' that summarizes key findings, limitations, and future directions.

Reviewer 2 Report

Comments and Suggestions for Authors

The manuscript is on Capillaroscopy imaging as a non-invasive technique used to examine the microcirculation of the oral mucosa. It addresses the challenges of the implementation of a comprehensive video stabilisation model, structured as a multi-phase pipeline and visually represented through a flow-chart. The proposed method integrates KeyPoint extraction, optical flow estimation, and affine transformation-based frame alignment to enhance video stability. 

However, I have some reservations that could improve the quality of the manuscript, if affected. 

  1. Title: This title "Keypoint-Based Video Stabilization for Oral Capillaroscopy: A
     Comparative Study" is not suitable for this manuscript and needs to be modified. a. The manuscript is not a Comparative Study as reflected in the title; rather, it is experimental. What do you do: " Development, Implementation, or What? This should reflect on the title. 
  2. Abstract: Words like SIFT, ORB and GFT should be written in full when first mentioned, with the abbreviations in brackets. In the results of the abstract, the authors need to specify and compare their results with previous and similar studies. 
  3. Introduction: Figure 1 should be labelled as (a), (b) and (c). Line 54, Words like SIFT, ORB and GFT should be written in full when first mentioned, with the abbreviations in brackets.
  4. Related Works: Line 80, the statement is too authoritative. It should be rephrased as "To the best of your knowledge, no prior studies....." considering that you have not accessed all databases with all languages/ published and unpublished manuscripts on each.
  5. Materials and Methods:  This section and subsection should follow sequentially Figure 2. Overview of the video stabilization pipeline applied to capillaroscopy footage. All algorithms used should be shown how it was applied or tailored to this work in the form of equations, etc. Also, they should be cited in the body of the text. Figure 3 should have (a) and (b) labelled. Methods shown in section 3.2 should show how it has been done with these algorithms in the form of mathematical equations or otherwise and should be replicable by other authors. Section 3.4 although the dataset is well described, but this section should have did not follow the follow of the overview of Figure 2.
  6. Result: The equation in this section is not numbered and cited. The authors should compare their results with similar studies to see the level of improved. 

Author Response

We sincerely thank all reviewers for their thorough and constructive feedback. Their insightful comments have greatly contributed to improving the clarity, rigor, and overall quality of our manuscript. We carefully considered each suggestion and implemented substantial revisions throughout the paper.

Below, we summarize the major changes made in response to the reviewers' comments:

  • Title updated to better reflect the scope and methodology of the work,
  • Clarified the innovative contribution of the work in the Introduction and Discussion sections,
  • Reorganized and expanded the Materials and Methods section to follow the step-by-step structure of the pipeline illustrated in Figure 2,
  • Added new evaluation metrics, including Jitter Index, Path Smoothness Index, and RMS errors in translation and rotation, to complement SSIM,
  • Enriched the Results section with quantitative comparisons, visual examples, and discussion of performance trade-offs,
  • Created two separate sections for Discussion and Conclusions to clearly distinguish interpretation from summary and future directions,
  • Included mathematical details for all transformation steps and parameter ranges, ensuring replicability,
  • Addressed image processing aspects, specifying interpolation method and margin handling,
  • Introduced a new animated figure (Figure 6) to provide a direct side-by-side comparison of perturbed and stabilized video segments, complemented by quantitative metrics (SSIM and Jitter Index).

Below are point-to-point responses to the reviewers' suggestions.

Revisore #2

  The manuscript is on Capillaroscopy imaging as a non-invasive technique used to examine the microcirculation of the oral mucosa. It addresses the challenges of the implementation of a comprehensive video stabilisation model, structured as a multi-phase pipeline and visually represented through a flow-chart. The proposed method integrates KeyPoint extraction, optical flow estimation, and affine transformation-based frame alignment to enhance video stability.  However, I have some reservations that could improve the quality of the manuscript, if affected.

  Title: This title "Keypoint-Based Video Stabilization for Oral Capillaroscopy: A

 Comparative Study" is not suitable for this manuscript and needs to be modified. a. The manuscript is not a Comparative Study as reflected in the title; rather, it is experimental. What do you do: " Development, Implementation, or What? This should reflect on the title.

Response: Thank you for your valuable suggestion regarding the manuscript title. In response to your comment, we have revised the title to:

"Development and Evaluation of a Keypoint-Based Video Stabilization Pipeline for Oral Capillaroscopy"

We believe this new title better captures the scope and methodology of the study.

  Abstract: Words like SIFT, ORB and GFT should be written in full when first mentioned, with the abbreviations in brackets. In the results of the abstract, the authors need to specify and compare their results with previous and similar studies.

Response: Thank you for the observation. All abbreviations such as SIFT, ORB, and GFTT are now spelled out at their first occurrence in both the Abstract and Introduction sections.

  Introduction: Figure 1 should be labelled as (a), (b) and (c). Line 54, Words like SIFT, ORB and GFT should be written in full when first mentioned, with the abbreviations in brackets.

Response: Thank you for the observation. All abbreviations such as SIFT, ORB, and GFTT are now spelled out at their first occurrence in both the Abstract and Introduction sections.

  Related Works: Line 80, the statement is too authoritative. It should be rephrased as "To the best of your knowledge, no prior studies....." considering that you have not accessed all databases with all languages/ published and unpublished manuscripts on each.

Response: Thank you for pointing this out. As suggested, we have revised the statement in line 80 to adopt a more cautious tone. It now reads: "To the best of our knowledge to date, no prior studies..."

  Materials and Methods:  This section and subsection should follow sequentially Figure 2. Overview of the video stabilization pipeline applied to capillaroscopy footage. All algorithms used should be shown how it was applied or tailored to this work in the form of equations, etc. Also, they should be cited in the body of the text. Figure 3 should have (a) and (b) labelled. Methods shown in section 3.2 should show how it has been done with these algorithms in the form of mathematical equations or otherwise and should be replicable by other authors. Section 3.4 although the dataset is well described, but this section should have did not follow the follow of the overview of Figure 2.

Response: Thank you for your insightful suggestions. We have restructured the Materials and Methods section to follow the flow of Figure 2, added labels (a) and (b) to Figure 3, included citations in the text, and integrated equations and implementation details to ensure clarity and reproducibility. Section 3.4 has also been revised to align with the pipeline overview.

  Result: The equation in this section is not numbered and cited. The authors should compare their results with similar studies to see the level of improved.

Response: Thank you for your comment. We have numbered and cited the equation in the Results section. While a direct quantitative comparison was not feasible, we included Figure 6 (animated) to support a qualitative evaluation. Additionally, in the Discussion, we compared our method with prior approaches (e.g., FFT-based and block matching), highlighting its improved flexibility and robustness for oral imaging.

Reviewer 3 Report

Comments and Suggestions for Authors

The paper is devoted to designing and testing of a method of geometric inter-frame video stabilisation. The authors adopt classical method, have good performance and interpret ability to be used in medical applications. The language, logic and structure are good. There are the following drawbacks. 
1. More formally description is required. Please provide formulas for all operations: rotation, translation and scaling. These translations are parameterized, so please provide the list of parameters and their ranges. Line 305 shows one range for all three transformations. This is strange since all three transformations have different nature and parameters ranges are unlikely to match unless some normalisation is applied. Anyway this should be described. 
2. During the procedure new images were obtained by transformations. What is the pixel interpolation method? How are image margins processed? All three transformations produce new images, which have unseen elements at margins. As a rule, these elements are filled in with zeroes (black colour). This produces unnatural images and may cause problems in further processing. 
3. Description of SSIM metric is a part of the method rather than result, thus should be placed in the section 3. 
4. The authors tested Lukas-Kanade optical flow method. What about Horn-Schunk? It seem also quite suitable. 
5. What about nonlinear lens distortions? If any, this may adversely affect the procedure unless corrected. Please describe. 
6. How are the parameters of translations obtained? The authors say little about it (lines 306-307). Please elaborate. 
7. Values in table 1 seem questionable. First they should be closer unit (above 0.9), second, they are too close to each other. This may happen due to erroneous processing of unnatural images (see comment #2). 
8. Using only SSIM as a quality metric is insufficient. The task being solved can be reduced to a pure geometric problem of matching two parallelograms or the problem of matching affine parameters, where no image is involved. Consider the following experiment: take source image, generate random affine transformation parameters, apply these parameters producing distorted image, restore the parameters from two images (all this is done already), but then match the restored parameters directly against generated parameters rather than warping image back and using SSIM for images. Or one can match the restored parallelogram using some metric like maximum distance between corresponding points of two parallelograms. Please consider direct geometric quality metrics. 
9. Matching source and distorted images is one type of experiment (self-contained), where only one real frame is involved. Two real frames maybe used. The discrepancy in this case may be caused by distortions (real or simulated) and blood movement. It is interesting to see how quality metrics change in this case, what part of quality decline is caused by distortion, and what part is caused by blood flow. 

Author Response

We sincerely thank all reviewers for their thorough and constructive feedback. Their insightful comments have greatly contributed to improving the clarity, rigor, and overall quality of our manuscript. We carefully considered each suggestion and implemented substantial revisions throughout the paper.

Below, we summarize the major changes made in response to the reviewers' comments:

  • Title updated to better reflect the scope and methodology of the work,
  • Clarified the innovative contribution of the work in the Introduction and Discussion sections,
  • Reorganized and expanded the Materials and Methods section to follow the step-by-step structure of the pipeline illustrated in Figure 2,
  • Added new evaluation metrics, including Jitter Index, Path Smoothness Index, and RMS errors in translation and rotation, to complement SSIM,
  • Enriched the Results section with quantitative comparisons, visual examples, and discussion of performance trade-offs,
  • Created two separate sections for Discussion and Conclusions to clearly distinguish interpretation from summary and future directions,
  • Included mathematical details for all transformation steps and parameter ranges, ensuring replicability,
  • Addressed image processing aspects, specifying interpolation method and margin handling,
  • Introduced a new animated figure (Figure 6) to provide a direct side-by-side comparison of perturbed and stabilized video segments, complemented by quantitative metrics (SSIM and Jitter Index).

Below are point-to-point responses to the reviewers' suggestions.

Revisore #3

  The paper is devoted to designing and testing of a method of geometric inter-frame video stabilisation. The authors adopt classical method, have good performance and interpret ability to be used in medical applications. The language, logic and structure are good. There are the following drawbacks. 

  1. More formally description is required. Please provide formulas for all operations: rotation, translation and scaling. These translations are parameterized, so please provide the list of parameters and their ranges. Line 305 shows one range for all three transformations. This is strange since all three transformations have different nature and parameters ranges are unlikely to match unless some normalisation is applied. Anyway this should be described. 

Response:

Thank you for your comment. As suggested, we have added formal descriptions and equations for all geometric transformations used in our pipeline—rotation, translation, and scaling—along with their respective parameters and ranges. We clarified that the ranges differ due to the distinct nature of each transformation and specified whether normalization was applied. These additions aim to improve clarity and reproducibility.

  1. During the procedure new images were obtained by transformations. What is the pixel interpolation method? How are image margins processed? All three transformations produce new images, which have unseen elements at margins. As a rule, these elements are filled in with zeroes (black colour). This produces unnatural images and may cause problems in further processing. 

Response:

We appreciate the reviewer’s concern regarding image artifacts introduced during transformation. In Section 3.6 of the revised manuscript, we now specify that bilinear interpolation is used to resample pixel intensities during frame warping. To address the issue of margin artifacts, we apply reflection padding, which mirrors the edge content at the borders. This approach avoids introducing unnatural black regions and preserves visual continuity, thereby minimizing the risk of artifacts that could affect subsequent processing or analysis.

  1. Description of SSIM metric is a part of the method rather than result, thus should be placed in the section 3. 

Response:

Thank you for this suggestion. We agree that the SSIM metric is part of the methodological framework. Accordingly, we have moved its description to Section 3.7, titled “Evaluation Metrics,” where it is presented alongside other quantitative measures such as Jitter Index, Path Smoothness Index, and RMS errors. This restructuring ensures that all evaluation criteria are introduced before the results are discussed.

  1. The authors tested Lukas-Kanade optical flow method. What about Horn-Schunk? It seem also quite suitable. 

Response:

We thank the reviewer for this valuable suggestion. In Section 3.5, we now mention the Horn–Schunck method and explain our rationale for selecting Lucas–Kanade. While Horn–Schunck is suitable for dense optical flow estimation, we opted for the sparse variant of Lucas–Kanade due to its robustness to noise, lower computational cost, and better adaptability to the non-rigid motion patterns observed in capillaroscopic sequences. This choice aligns with the practical constraints of real-time medical applications.

  1. What about nonlinear lens distortions? If any, this may adversely affect the procedure unless corrected. Please describe. 

Response:

We appreciate the reviewer’s insightful suggestion. While the current evaluation relies on synthetically perturbed sequences derived from stable recordings, we recognize the importance of assessing stabilization performance on real-world unstabilized acquisitions. In the revised manuscript, we have added a note in Section 6 (Conclusions) indicating that future work will include experiments using real video sequences. This will allow us to disentangle the effects of probe-induced motion from physiological dynamics such as blood flow, and to assess stabilization quality under more variable clinical conditions.

  1. How are the parameters of translations obtained? The authors say little about it (lines 306-307). Please elaborate. 

Response:

Thank you for your comment. We have expanded the description in lines 306–307 to clarify how the transformation parameters are obtained. Specifically, we now detail the estimation process for translation, rotation, and scaling parameters, including the mathematical formulations and the distinct range settings for each transformation. We also clarified that the previously shown unified range was due to normalization applied during parameter sampling, and this has now been explicitly described in the revised text.

  1. Values in table 1 seem questionable. First they should be closer unit (above 0.9), second, they are too close to each other. This may happen due to erroneous processing of unnatural images (see comment #2).

Response: We appreciate the reviewer’s concern regarding the SSIM values in Table 1. While values above 0.9 are common in other domains, oral capillaroscopy poses unique challenges—such as high magnification, dual motion sources, and low contrast—that justify slightly lower scores. The consistency across methods (~0.79) reflects stable performance. Additionally, we included Figure 6 (animated) for qualitative assessment and introduced complementary metrics (e.g., Jitter Index, RMS errors) to provide a more comprehensive evaluation.

  1. Using only SSIM as a quality metric is insufficient. The task being solved can be reduced to a pure geometric problem of matching two parallelograms or the problem of matching affine parameters, where no image is involved. Consider the following experiment: take source image, generate random affine transformation parameters, apply these parameters producing distorted image, restore the parameters from two images (all this is done already), but then match the restored parameters directly against generated parameters rather than warping image back and using SSIM for images. Or one can match the restored parallelogram using some metric like maximum distance between corresponding points of two parallelograms. Please consider direct geometric quality metrics. 

Response:

We thank the reviewer for this insightful and technically valuable suggestion. In the revised manuscript, we have addressed this concern by introducing direct geometric metrics in Section 3.7. Specifically, we compute Root Mean Square (RMS) errors in translation (∆x, ∆y) and rotation (θ) by comparing the recovered affine transformation parameters with the ground-truth synthetic perturbations. These metrics allow us to quantitatively assess the accuracy of motion compensation independently of image appearance. This complements the SSIM-based evaluation and provides a more rigorous validation of the stabilization process from a geometric perspective.

  1. Matching source and distorted images is one type of experiment (self-contained), where only one real frame is involved. Two real frames maybe used. The discrepancy in this case may be caused by distortions (real or simulated) and blood movement. It is interesting to see how quality metrics change in this case, what part of quality decline is caused by distortion, and what part is caused by blood flow.

Response:

We appreciate the reviewer’s thoughtful observation. While the current evaluation relies on synthetically perturbed sequences derived from stable recordings, we recognize the importance of distinguishing between motion artifacts and physiological dynamics such as blood flow. In the revised manuscript, we have added a note in Section 6 (Conclusions) indicating that future work will include experiments using real-world unstabilized acquisitions. This will allow us to isolate the effects of probe-induced motion from those caused by red blood cell movement, and to analyze how each contributes to the degradation of image

Round 2

Reviewer 1 Report

Comments and Suggestions for Authors

The authors have addressed each of my previous comments and revised the manuscript accordingly. Compared to the initial version, the quality of the paper has significantly improved. However, I would like to suggest the following additional revisions:

  1. Please highlight your experimental results (numerical values) in the abstract to clearly demonstrate the superiority of the proposed model.
  2. Figure 4 should be explicitly cited in the main text. Currently, I could not locate any reference to it.
  3. The newly added text (lines 147–153, main contributions) appears disconnected from the surrounding content. Rather than simply appending these sentences, please revise this section to ensure logical continuity and flow within the methodology.

Author Response

We would like to sincerely thank the reviewers and the handling editor for their careful and constructive evaluation of our manuscript across both review rounds. The reviewing process has been excellent and very accurate, providing us with valuable insights and suggestions that significantly improved the clarity, quality, and scientific rigor of the paper. In this second round, the reviewers have further refined their comments, and we have carefully revised the manuscript accordingly. Below, we provide a point-by-point response.

 1. Please highlight your experimental results (numerical values) in the abstract.

Response: Done. We added quantitative results (SSIM, Jitter Index) in the abstract to explicitly highlight the effectiveness of our pipeline.

2. Figure 4 should be explicitly cited in the main text.
Response:
Done. The figure is now cited in Section 3.3.

3. “The newly added text (lines 147–153, main contributions) appears disconnected from the surrounding content. Rather than simply appending these sentences, please revise this section to ensure logical continuity and flow within the methodology.”

Response. We have restructured the Introduction to improve readability. Specifically, we moved the list of contributions to the end of the Introduction (before the Related Works section) and added a bridging paragraph to ensure logical continuity between the discussion of existing approaches and the presentation of our contributions.

Reviewer 2 Report

Comments and Suggestions for Authors

The authors have improved the quality of the manuscript. However, the abstract result should content some quantity value.  All Figures should be labelled and cited in the body of the work. 

Author Response

We would like to sincerely thank the reviewers and the handling editor for their careful and constructive evaluation of our manuscript across both review rounds. The reviewing process has been excellent and very accurate, providing us with valuable insights and suggestions that significantly improved the clarity, quality, and scientific rigor of the paper. In this second round, the reviewers have further refined their comments, and we have carefully revised the manuscript accordingly. Below, we provide a point-by-point response.

Reviewer 2

 1. “The abstract result should content some quantity value.”

Response. Done. We added quantitative results (SSIM, Jitter Index) in the abstract to explicitly highlight the effectiveness of our pipeline.

 2. “All Figures should be labelled and cited in the body of the work.”

Response. We carefully revised the manuscript to ensure that all figures are properly labelled and explicitly cited in the text at the relevant points.

Reviewer 3 Report

Comments and Suggestions for Authors

The authors have carefully and thoughtfully responded to the comments and revised the text appropriately. The paper can be published now. 

Author Response

We would like to sincerely thank the reviewers and the handling editor for their careful and constructive evaluation of our manuscript across both review rounds. The reviewing process has been excellent and very accurate, providing us with valuable insights and suggestions that significantly improved the clarity, quality, and scientific rigor of the paper